# Patient and physiotherapist perceptions of the Getting Recovery Right After Neck Dissection (GRRAND) rehabilitation intervention: a qualitative interview study embedded within a feasibility trial

Beth Fordham ![ORCID],[1] Toby O Smith,[1,2] Sarah Lamb,[1,3] Alana Morris,[1] Stuart C Winter[4]

[1]Nuffield Department of Orthopaedics, Rheumatology and Musculoskeletal Sciences, University of Oxford, Oxford, UK
[2]Faculty of Medicine and Health Sciences, University of East Anglia, Norwich, UK
[3]College of Medicine and Health, University of Exeter, Exeter, UK
[4]Nuffield Department of Surgical Sciences, University of Oxford, Oxford, UK

**Correspondence to**
Dr Beth Fordham;
beth.fordham@ndorms.ox.ac.uk

## ABSTRACT

**Objective** The Getting Recovery Right After Neck Dissection (GRRAND) intervention is a physiotherapy programme for patients with head and neck cancer who have undergone neck dissection. The aim of this qualitative study was to understand if the intervention was useful, acceptable and whether it was feasible to conduct a randomised controlled trial (RCT).

**Design** This qualitative study was embedded within the GRRAND-Feasibility (GRRAND-F) Study.

**Setting** Participants were recruited from four acute National Health Service hospitals in England between 2020 and 2021.

**Participants** We interviewed four usual care and four intervention patient-participants from a single study site (Oxford). Six were male, two were female. All were white British ethnicity. We interviewed two physiotherapists from Oxford who delivered the GRRAND-F intervention, and physiotherapists from Birmingham, Poole and Norwich who were trained to deliver the intervention but were not able to deliver it within the study time frame.

**Results** The analysis identified five themes: (1) Acceptability, (2) Adherence, (3) Outcomes, (4) Feasibility and (5) Stand-alone themes (prehabilitation, video consultations, healthcare use).

Patient-participants and physiotherapist-participants agreed that usual care was not meeting patients' rehabilitation needs. The GRRAND intervention provided biopsychosocial support. In comparison to the usual care group, patient-participants who received the intervention were more confident that they could perform rehabilitation exercises and were more motivated to engage in long-term adaptive behaviour change. Physiotherapists felt they needed more administrative support to participate in an RCT.

**Conclusion** Participants felt that usual care was insufficient. GRRAND provided much needed, biopsychosocial support to patients. Participants were supportive that it would be feasible to test GRRAND in an RCT.

**Trial registration number** ISRCTN11979997.

## STRENGTHS AND LIMITATIONS OF THIS STUDY

⇒ Immersive qualitative methodology. One researcher performed all the interviews and all primary analysis.
⇒ The analysis was iterative. The qualitative researcher presented the primary analysis to a multidisciplinary trial management group, which included surgeons, patients, physiotherapists and methodologists in order to refine, refute and further explore in subsequent interviews.
⇒ The sample was much smaller and homogeneous than anticipated due to the impact of COVID-19 restrictions on the trial recruitment. Consequently, the generalisability of the study findings is restricted.
⇒ We interviewed patient-participants and physiotherapist-participants and triangulated their data. This identified discrepancies between the two perspectives.

## INTRODUCTION

Head and neck cancer (HNC) incidence continues to rise and is anticipated to reach 1.08 million cases annually by 2030.[1] Survival rates for people living with this condition have also improved.[2] Consequently, there is a growing population of people living with HNC.

The clinical presentations of HNC are varied, reflected in a diversity in treatment options and treatment side effects. Treatment can involve surgery or radiotherapy (chemoradiotherapy) to the primary site and to the neck. However despite changing treatment paradigms up to 20% of patients with HNC still require a neck dissection (ND) as part of their treatment.[3] During the initial five postoperative years, patients may experience a range of side effects on a continuum of severity. The physical health side effects

can include problems with swallowing, speech, neck and shoulder function, sleeping, cervical contracture and muscle wastage.[4] The psychosocial side effects can include prolonged fatigue, depression, anxiety and isolation.[5] Fifty per cent of patients with HNC who have undergone ND cease to work due to their treatment side effects.[6]

Cochrane[7] and the National Institute for Health and Care Excellence[8] have highlighted the need for research into developing an intervention to promote recovery post-ND for patients with HNC. The Getting Recovery Right After Neck Dissection (GRRAND) Feasibility Study[9] aimed to assess the feasibility of conducting a randomised controlled trial (RCT) examining the clinical effectiveness and cost-effectiveness of a multimodal rehabilitation intervention. Embedded in this feasibility trial, we conducted a qualitative investigation to understand and represent participant and physiotherapist experiences of the GRRAND intervention.

The aim of this qualitative study was to understand the perspectives of the physiotherapists and patient participants on whether the GRRAND intervention was acceptable and/or perceived as useful and if it was feasible to conduct an RCT. This study has implications beyond physiotherapy following ND as there is increasing research into using personalised treatment approaches which are adapted in response to each patient's individual needs (adaptive therapy). This study will provide further insight into the barriers and feasibility of such regimes.

## METHODS
### Theoretical framework
We adopted a qualitative methodology to achieve a clearer picture of the GRRAND-Feasibility (GRRAND-F) process under investigation.[10] The qualitative methodology enables us to provide an in-depth interpretation of the quantitative data.[11] The study is reported in line with the Standards for Reporting Qualitative Research (checklist available).[12]

### Design and setting
This qualitative study was embedded within the GRRAND-F Study. From 2020 to 2021 participants recruited at the University of Oxford Hospital were randomised 2:1 to either receive the GRRAND intervention or to usual care.

### Interviews
Semistructured interviews were conducted to enable the interviewer to direct the conversations to answer the study's research questions while allowing the participants to freely represent their perspectives and introduce novel themes.[13]

Our Patient and public involvement (PPI) representatives and members of a patient support group (Heads-2gether) suggested that some participants might be inhibited to express in a group setting their genuine experiences because there was a large variation in distress and disability. Therefore, we used individual interviews

as opposed to focus groups,[14] in order that we did not inhibit participant disclosure in a focus group.

We did not conduct any repeat interviews nor did we return the transcripts to participants for member checking because we wanted to avoid overburden on a vulnerable patient group.

### Semistructured interview procedure and topic guides
A health psychology researcher (BF) (PhD, female) with over 10 years of experience designing, conducting and reporting qualitative research, conducted the interviews and analysis. The interviews were intended to be conducted face to face. However due to COVID-19 restrictions, we adapted the procedure and conducted all interviews over Microsoft Teams. Participants who consented to participate in the interviews were contacted by phone by BF to ensure continued consent despite conducting the interviews via videoconference technology. At the beginning of the interview, BF took time to explain to the participants that she was a psychologist who wanted to understand their experience of participating in this feasibility trial (see semi-structured interview (SSI) guide); that she was not a physiotherapist nor a leading member of the trial design team, who might have hopes to prove GRRAND was useful; BF was not aiming to prove one hypothesis or another; and she reiterated that the participant's data would remain anonymous to all other members of the physiotherapy and research teams.

The interviews lasted between 30 min and 60 min. We interviewed all patient participants who had completed 6 months of participation in either the treatment as usual or the GRRAND intervention arm. The interviews were recorded on Microsoft Teams and the audio (mp4) files downloaded. The audio files were uploaded via a secure server to a transcription service and the anonymous transcriptions were returned, via secure server to BF.

The semistructured interview topic guide (table 1) was developed from a scoping review of the qualitative literature of the lived experiences of patients with HNC.[15–19] The guide was revised by the GRRAND-F Trial management group which included PPI, surgeon, physiotherapist and psychologist perspectives.

### Participants
#### Sampling and recruitment
Patients with HNC who had undergone ND and consented to be part of the GRRAND-F trial were invited to participate in individual video interviews 6 months after their ND. Our original protocol planned to purposively sample from the broader GRRAND-F participant sample on the basis of sex, ethnicity and trial site. However, due to complications from the COVID-19 lockdown, our trial sample size was greatly reduced. Therefore, we revised our methodology to invite all the patient-participants who had participated in the GRRAND-F Study through a convenience rather than purposive sampling strategy. We invited patient-participants who withdrew from GRRAND-F, but all declined (n=4). All physiotherapists

**Table 1** Semistructured patient-participant interview guide

| SSI objective | Content |
|---|---|
| Introduction and rapport build | No right or wrong answers, take your time we want to learn as much as we can from you. You are the experts. |
| Feasibility | |
| Approach | Do you remember at what point you were approached about being part of this study? |
| Participation | Can you tell me what you first thought about participating in a study like this? |
| Randomisation | When you were approached about the study, were told that you might receive one type of programme or you might receive a different type? Can you tell me about these options? |
| Questionnaires | You completed a set of questionnaires before and after completing the GRRAND-F programme. What did you think about these questions? (*Share the questionnaires to remind if nothing is remembered*). |
| Acceptability and usefulness | |
| Treatment as usual | When you were discharged from hospital, were you given a booklet of physiotherapy exercises to take home with you? *Here is a copy - Show example*. What did you think about the physiotherapy care you received while you were in hospital? |
| GRRAND intervention | You have received X (eg, 3) sessions of physiotherapy since your operation in X (eg, September), can you tell me what these sessions were like? |
| Delivery | Can you tell me, were your appointments delivered via video calls, or face to face or a mixture of both? |
| Attendance | Were there any sessions which you were unable to attend? Can you remember why you were unable to attend? Is there anything which the physiotherapy team could have done to make it easier for you to attend? Can you tell me about why you were not able to attend some sessions? |
| GRRAND usefulness | Did you think the physiotherapy sessions have helped you recover after your operation? |
| GRRAND session content | Can you identify any specific parts of the sessions which stood out for you? Parts which really helped? Parts you struggled with? And parts you did not understand why you were doing them? |
| GRRAND homework content | Did the physio give you an exercise diary and/or a printed set of physiotherapy for you to complete at home? (show examples) Can you remember what you received? |
| External services | Have you sought any other type of help during your rehabilitation outside of what we have offered you in this trial? |
| Free feedback, summary and close | Do you have any other feedback you would like to talk about? Any questions. Summarise discussion points and close. |

GRRAND, Getting Recovery Right After Neck Dissection.

who participated in the trial were invited and agreed to participate in the interviews. All patient-participants and physiotherapist-participants provided informed consent.

## Usual care and GRRAND-rehabilitation intervention description

A more detailed account of the GRRAND-F intervention and trial is presented in the study protocol. Gallyer *et al*[9] In brief, both usual care and GRRAND intervention participants received advice on performing a range of movement exercise, respiratory care, body position, oral health, pain management and pacing activities. They also received a booklet outlining self-management techniques including exercise, pain management, return to work and activities of daily living.

The GRRAND intervention participants receive an individualised rehabilitation programme delivered by outpatient physiotherapists who have been specifically trained. They will also receive a home exercise programme to supplement the face-to-face sessions.

## Data analysis

The qualitative data were managed using NVIVO software and analysed using Framework Analysis[20] by a single researcher (BF). We built a thematic framework in advance regarding the feasibility of the trial based on

previous trials regarding the feasibility of a novel intervention.[21] We built a second framework regarding the participants' acceptability and adherence to the GRRAND-F intervention based on the capability, opportunity, motivation theory of behaviour change (COM-B).[22] The COM-B model was used to understand why participants did or did not use their prescribed physiotherapy exercises.

BF began familiarisation with the data while conducting the interviews and recorded reflections and notes after each interview. BF then read the transcriptions of the interviews and began building a preliminary framework of themes with supporting quotations. Preliminary themes were iteratively presented, discussed and refined with the trial management group as part of the analysis process. Deviant themes were identified and either redefined or represented as stand-alone themes. In our final trial management meeting we agreed data saturation had been met.

## Patient and public involvement
The GRRAND Trial management group included a patient representative. BF attended HNC patient-support group (heads2gether) meetings to contextualise the analysis and present the developing analytical framework.

## RESULTS
### Participants
We interviewed four usual care and four intervention patient-participants between March and November 2021. Two participants were invited but declined to participate in the interview study. All were recruited from the Oxford site. Six of the eight were male and two were female. All were white British ethnicity. Six were in paid work and two were out of work. All patients had undergone an ND as part of their treatment plan. All treatments were curative intent and all patients were clinically cancer-free at the time of interview

We interviewed two physiotherapists from Oxford and one from Norwich who delivered the GRRAND-F intervention. In addition, we interviewed one physiotherapist from Birmingham and one from Poole who were trained in the intervention and study processes but had not been able to deliver the intervention due to COVID-19 restrictions.

Table 2 reports the themes and quotes from participants regarding their experiences of the GRRAND-F Trial. The patient-participant evidence quotations are labelled as participant ID and the intervention arm they were allocated to, that is, usual care or intervention arm. For the physiotherapists, we labelled them as a physiotherapist and the location they work in.

## Patient and physiotherapist themes
The overarching coding tree, generated in NVIVO, is presented in table 2. The evidence quotes' references throughout the results are presented in tables 3 and 4.

**Table 2** Coding tree

| Main theme | Subtheme | Summary |
|---|---|---|
| Acceptability | Complexity of needs | Variation and lack of correlation |
| | Physiotherapy content | Novel |
| | | Standard care elements |
| | | Not all useful |
| | Psychosocial content | Fear avoidance |
| | | Pacing |
| | | Reassurance |
| | | Emotional support |
| | | Empowerment and personal control |
| | Usual care comparison | Inpatient care is not enough |
| Usefulness | Adherence | Capability |
| | | Opportunity |
| | | Motivation |
| | Outcomes | Physical |
| | | Psychological |
| | | Social |
| Standalone themes | Prehabilitation | Understanding more about what to expect |
| | Video consultation | Hybrid due to some benefits and some problems |
| | Additional care | None used |
| Feasibility | Recruitment | High motivation for engagement |
| | Randomisation | Acceptable |
| | Questionnaires | Mental health misunderstood for some |

## Theme 1: Acceptability
### Complexity of care needs
When exploring the acceptability of the GRRAND intervention it was important to recognise the variation of patient-participant experiences. Each patient-participant experienced different physical symptoms with a range of severity (*Quotes 1 and 2*). Some patient-participants reported disabling psychological symptoms (Quotes 3 and 4), and, unhelpful avoidant social coping strategies (*Quotes 5 and 6*).

The physiotherapist-participants recognised that the psychosocial distress experienced by these patients is not correlated to the physical symptom type or severity (*Quote 7*).

Consequently, we understood that each patient experiences a unique set of physical, psychological and social symptoms and will likely hold varying expectations and

| Table 3 | Evidence quotes for theme 1 |
|---|---|
| **Quote** | **Evidence quotation** |
| Theme 1: Acceptability of the GRRAND intervention | |
| 1 | *"So, my main problem at the moment is neck mobility in as far as I can't look over my shoulders…"* (Participant 114 Usual care) |
| 2 | *"I've not been in any pain, so I'm very lucky"* (Participant 109 Intervention) |
| 3 | *"I've found the mental challenge 100 times more challenging than the physical type"* (Participant 109 Intervention) |
| 4 | *"I did not know or realise how I would be permanently damaged with my mind,…"* (Participant 102 Usual care) |
| 5 | *"The neighbour invited me for a Christmas Eve drink, and I don't drink … in front of anybody because I slurp, so I thought no, I'll be fine, and I just started slurping and I thought I can't even go out and have a face-to-face drink with the neighbour."* (Participant 102 Usual care) |
| 6 | *"They worry about how they look, they don't want to go and meet their friends because they worry about how they appear. …"* (Physiotherapist Oxford 2) |
| 7 | *"Two people can have the same cancer and the same treatment and they could react to treatment very differently but even if they had exactly the same symptoms, the impact of that would be really different."* (Physiotherapist Oxford 1) |
| 8 | *"I think some of them really are fine and are functioning really well, even if they have got a problem, and therefore there is a danger that we can really over-treat them, and I suppose there is the potential that GRRAND exacerbates that…"* (Physiotherapist Norwich 1) |
| 9 | *"I think the training would need to be delivered, ideally I think by somebody with a lot of clinical experience in that area."* (Physiotherapist Oxford 1) |
| 10 | *"I think some patients hit you harder than others, I think I just want to make sure I'm giving them the support they need and I found I've reflected a lot on is 'have I been able to give them the right amount of support, have I said the right thing at the right time?'"* (Physiotherapist Oxford 2) |
| 11 | *"…hopefully we can pick up people around the country through the trial and have our own network … we all have different thoughts and different views of managing things…"* (Physiotherapist Norwich 1) |
| 12 | *"Right, the first things that helped, definitely the stretching of my neck from side to side, my head from side to side. I'm still very tight and I'm still very stiff at times, so I push, but that's a lot better than it was, which means that I can drive."* (Participant 107 Intervention) |
| 13 | *"It's made me do a bit more with them. As I said, we always followed them up; we always did things. I didn't used to do so much of the resistance work, which I know more and more evidence has been coming out to suggest that, so I'm tentatively doing that a bit more."* (Physiotherapist Norwich 1) |
| 14 | "Well just those ones I showed you at the back, which was pointless… they weren't really doing much." (Participant 106 Intervention) |
| 15 | *"Our surgeons don't like TMJ resisted exercises, so we definitely don't use those. When he read the booklet, he was horrified.'"* (Physiotherapist Norwich 1) |
| 16 | *"Then the exercise component I think is always really helpful, that people know what they're doing and how confident they can be handling their arm and handling their scars and that kind of thing."* (Physiotherapist Oxford 1) |
| 17 | *"I think pacing is huge. It's something we use a lot in physiotherapy, to be fair, anyway, so that at least wasn't so new to me."* (Physiotherapist Norwich 1) |
| 18 | *"I would be more and more worried, thinking, 'What is this? Why is this happening?' and exaggerating in my head, thinking it's far worse than it actually is.*<br>So, it's for reassurance (Interviewer) *Yes."* (Participant 107 Intervention) |
| 19 | *"I think education is often overlooked as quite a small section, but I think it's absolutely huge. You can have some sessions where you just talk to patients."* (Physiotherapist Oxford 2) |
| 20 | *"I have to be careful because I could start going on and on about this and that and problems, and it is like a shoulder to cry on, as such."* (Participant 107 Intervention) |
| 21 | *"The patients have so much more to talk about than just their neck or their shoulder problem. Emotionally it's quite difficult and I've really have seen a lot of that actually."* (Physiotherapist Oxford 2) |
| 22 | *"So, I think there is a really important psychological support aspect to it… validating concerns and … giving people an aspect of their care that they can actually influence too…because there are lots of things for the treatment, they just have to show up for and be obedient and accept it."* (Physiotherapist Oxford 1) |
| 23 | *"I know I definitely saw them before I left. I can't remember whether I was given a booklet or not. I'm sorry…"* (Participant 107 Intervention) |

Continued

| Table 3 | Continued |
|---------|-----------|
| **Quote** | **Evidence quotation** |
| 24 | *"As an inpatient … they want whatever it takes for them to get home because they don't want to be in hospital."* (Physiotherapist Oxford 1) |
| 25 | *"No when I was in the hospital I received physiotherapy. It was when I went home the problems started."* (Participant 114 Usual care) |

GRRAND, Getting Recovery Right After Neck Dissection; TMJ, temporomandibular joint.

need flexibility in their degree of support from the rehabilitation package.

The physiotherapist-participants highlighted the unmet psychosocial needs of their patients. They explained how treatment can be very different for each participant due to their variation in symptoms (*Quote 8*).

Due to the complexity of this patient population's symptom presentation, the physiotherapist-participants identified three future training needs in order to deliver GRRAND. One, a specialist HNC ND physiotherapist delivers the training (*Quote 9*). Two, receive more training on how to manage the psychosocial needs of the patient population (*Quote 10*). And, three, develop a communication network across HNC physiotherapists in order to deepen their shared learning (*Quote 11*).

### Physiotherapy content

When reflecting on the physiotherapy content of the GRRAND intervention, the patient-participants and physiotherapist-participants identified effective physiotherapeutic content (*Quotes 12 and 13*). However, some patient-participants felt there was content that was not relevant for them (*Quote 14*). And, one physiotherapist-participant reported how they did not use some of the content because their service did not agree with it (Quote 15).

### Psychosocial content

The patient-participants and physiotherapist-participants identified four psychosocial constructs which helped the patient-participants during their rehabilitation. (1) The reduction of fear-avoidance (*Quote 16*). (2) The use of pacing exercises, a physiotherapist-participant reflected that pacing was already commonly used in their clinical practice (*Quote 17*). (3) The patient-participants explained that they needed reassurance (*Quote 18*) and the physiotherapist-participants recognised how important their reassurance was for their patients (*Quote 19*). (4) All participants highlighted how the GRRAND intervention sessions were used as emotional support for the patient-participants (*Quotes 20 and 21*). It was also recognised that the physiotherapists could offer empowerment and personal control to patients who were feeling disempowered and overwhelmed (*Quote 22*).

### Inpatient care

The patient-participants in the usual care and intervention groups and the physiotherapist-participants, all recognised that the work they can do during inpatient care is limited because the patient is overwhelmed and highly motivated to get home rather than engaging with the physiotherapeutic work (*Quotes 23 and 24*). They also recognised the impact of the surgery on their daily functioning only becomes evident once they have settled back into home life (*Quote 25*).

### Theme 2: Behavioural adherence to GRRAND
#### Capability
Irrespective of whether the patient-participants experienced minor or major physical dysfunction, they reported feeling physically capable to perform the exercises prescribed to them after surgery, that is, they believed they were capable of doing them. Patient-participants in the intervention group, however, reported more psychological capability that is, they believed they can and they should perform the exercises (*Quote 26*). The usual care group patient-participants reported concerns that they were performing the exercises incorrectly and felt 'overwhelmed' to persist with the exercises in isolation (*Quote 27*).

#### Opportunity
All the patient-participants explained that they had social support which gave them opportunity to perform their GRRAND exercises (*Quote 28*). None identified any physical barriers which prevented them from performing the exercises (*Quote 29*). The physiotherapy-participants suggested the geographical regions where they worked tended to present fewer physical barriers for patients to perform their exercises at home (*Quote 30*).

#### Motivation
The motivation to perform the physiotherapy exercises appeared more varied across our interview sample. Importantly both members of the intervention group, the usual care group and the physiotherapists felt that regular sessions with a physiotherapist would increase their motivation to adhere to their exercise prescriptions (*Quote 31*). Physiotherapist-participants felt it was very important to tailor the intervention to meet the patient's individual goals (*Quote 32*).

One participant in the intervention group reflected that from discussions with physiotherapists and nurses, he began to understand that his rehabilitation post-ND surgery would be a lifelong process, that symptoms could become worse or new symptoms could arise across his life.

**Table 4** Evidence quotes for Themes 2–5

| | |
|---|---|
| **Theme 2: Behavioural adherence to GRRAND** | |
| 26 | *"…I was told to just push my fingers until there was resistance… not to go over the top, and they have definitely helped."* (Participant 107 Intervention) |
| 27 | *"I'm doing all this on my own and for the exercises I found that just a bit overwhelming."* (Participant 102 Usual care) |
| 28 | *"I'd tell the missus and the kids, and my boy, he's a professional footballer, so that side of it is very much a part of his life, so he had been nagging…come on dad, we're going out for a walk even if we go halfway, or something like that, so that was all very positive."* (Participant 109 Intervention) |
| 29 | *"Plenty of space and plenty of time, yeah. There was nothing at home to stop me from doing everything or anything. No, nothing."* (Participant 107 Intervention) |
| 30 | *"they've never said, 'I can't do them.'… we tend to have patients who live in nice expensive bungalows. ((City)) is a very different demographic."* (Physiotherapist Norwich 1) |
| 31 | *"…patients have said to me that they like having that check-in with the physios …They're doing their exercises, they're noticing no improvements…they think 'what's the point?' They touch base with us and it gives them that… (Motivation)… to carry on."* (Physiotherapist Oxford 2) |
| 32 | *"One gentleman wanted to go back to driving and he couldn't see his blind spot, so we really put the focus on the range of movement exercises and stretching to enable him to do that… So, it's really working towards their goals I suppose."* (Physiotherapist Oxford 2) |
| 33 | *"…it's important to keep doing it. The nurse last week said we call it, the afterlife treatment because it never seems to go away and you can go six months with no problems and then suddenly you start stiffening up, so it's important to keep doing them, probably for the rest of my life."* (Participant 109 Intervention) |
| **Theme 3: Patients' and physiotherapists' perceptions of clinical outcomes** | |
| 34 | *"How's it been helping? (laughs) It's helped me because if I didn't do it, it would be worse!"* (Participant 115 Intervention) |
| 35 | *"Yeah, because my neck, I can move quite freely now and I can lift my arms quite a long way up now, I can almost get it vertically now, I couldn't do that to begin with, but I can do it now. It's not easy, but I can do it."* (Participant 111 Usual care) |
| 36 | *"I don't believe it's helped with the swallowing. In fact, I think my swallowing did get better and now it's starting to get worse"* (Participant 107 Intervention) |
| 37 | *"… has the physio exercises helped with the swallowing? (Interviewer) No that's still about the same."* (Participant 108 Usual care) |
| 38 | *"I don't do anything with a swallow, because I don't want to cross over with them…they do them with SALT, so we don't do those."* (Physiotherapist Norwich 1) |
| 39 | *"Yeah, I think they have helped, definitely…. but I do think I've still got a long way to go."* (Participant 107 Intervention) |
| 40 | *"But there does come a point when you start flipping towards, it's not gonna get better."* (Physiotherapist Norwich 1) |
| 41 | *"At the moment, nothing's really helping for me to do anything more, I don't believe."* (Participant 107 Intervention) |
| **Theme 4: Stand-alone themes** | |
| 42 | *"What I didn't appreciate at the time was how much effect the dissection would have on me…I prefer…to know more about - certainly side effects were…"* (Participant 114 Usual care) |
| 43 | *"…before I had the operation, it could have been a good thing to have some physio beforehand…getting you something to focus on…and things, just maybe have something, so that when you're feeling low or something, it could give you something to aim at, focus on."* (Participant 107 Intervention) |
| 44 | *"So, I think it could be helpful to meet people beforehand but also we'd be giving them really vague information, like, well you might not have any symptoms at all or you might have this."* (Physiotherapist Oxford 1) |
| 45 | *"…in terms of conditioning, we could have a two, three week window of, 'Here's some exercises to get you fitter, to get your neck, your shoulder fitter, which is gonna help after your surgery and cause less restrictive range of movement.'"* (Physiotherapist Birmingham 1) |
| 46 | *"I don't think there is much to be achieved from doing it online, personally. I really don't."* (Participant 107 Intervention) |
| 47 | *"And now that we can actually get face to face appointments, still life gets in the way and it's difficult isn't it?"* (Carer for Participant 115 Intervention) |
| 48 | *"We did video because we needed to and alternating is okay. I didn't do any first assessments over video, they were all face to face which is where a lot of the really helpful stuff happens."* (Physiotherapist Oxford 1) |
| 49 | *"I had joined heads2gether"* (Participant 102 Usual care) |

Continued

| Table 4 | Continued |
|---|---|
| **Theme 2: Behavioural adherence to GRRAND** | |
| **Theme 5: Feasibility** | |
| 50 | *"I said: well yeah, if anything it will help even me or others, yeah I'm quite prepared to go ahead with it."* (Participant 108 Usual Care) |
| 51 | *"It was something I was really keen to be part of … I was very keen to keep involved and to deliver the intervention … the unmet need in this population is so profound."* (Physiotherapist Oxford 1) |
| 52 | *"So, it's taking time out of the day to look at when their appointment is, sometimes it's not there yet, you have to check back another day. It's all these little things that you're having to…, if this were to be a full-blown trial I think there'd need to be admin support definitely."* (Physiotherapist Oxford 2) |
| 53 | *"… I was actually quite disappointed. I think it was at that stage I realised how things would affect me in the long-term. I think it was around then I was told that I didn't qualify."* (Participant 102 Usual Care) |
| 54 | *"…we living in {County Name} we wouldn't be selected because the trials would be done in Oxford."* (Participant 108 Usual care) |
| 55 | *"I was there when she told him that he'd been put into the usual care group and he was absolutely gutted…So, I spent quite a lot of that session reassuring him actually, 'usual care is not a bad thing and you're still going to be getting physio."* (Physiotherapist Oxford 2) |
| 56 | *"Well, some of them I thought they were a waste of time … you know, your moods… and you were getting fed up of answering them as the page went on and on."* (Participant 108 Usual care) |
| 57 | *"Over the year his dysfunction worsened and worsened and worsened…"* (Physiotherapist Oxford 1) |

GRRAND, Getting Recovery Right After Neck Dissection; SALT, Speech and Language Therapist.

He believed that he could integrate his physiotherapy exercise routine into his daily life for a longer-term behaviour change (*Quote 33*). This could be an important information exchange whereby the physiotherapists encourage patients to identify exercises which they could keep doing for the rest of their lives to prevent future disability. This is an important consideration especially in finding daily life physical activities which can target the specific exercises objectives rather than a person having to repeat the same exercises forever.

### Theme 3: Patients' and physiotherapists' perceptions of clinical outcomes

Our participants felt that both the usual care and GRRAND intervention helped their rehabilitation (*Quote 34*) and specifically increased their physical functioning (*Quote 35*).

The main symptom which patient-participants did not feel was helped was their swallowing impairment (*Quotes 36 and 37*). One physiotherapist-participant explained swallowing exercises were covered by the Speech and Language Therapists and not by the physiotherapist (*Quote 38*).

Patient-participants from the intervention group recognised that their rehabilitation journey was going to be a long-term change to their life (*Quote 39*). However, some patient-participants and physiotherapist-participants recognised that they might not ever regain full function (*Quotes 40 and 41*).

### Theme 4: Stand-alone themes extra to feasibility and acceptability themes

Some patient-participants and physiotherapist-participants raised the idea of patient receiving support before their ND surgery that is, 'pre-habilitation'. They suggested patients could receive presurgery counselling to develop patient education regarding the spectrum of symptoms, symptom severity and comorbidities which they could experience (*Quote 42*). One patient-participant explained how he would have liked to have learnt some of the exercises preoperatively so that he could have become accustomed to them and felt prepared for the rehabilitation journey (*Quote 43*). Some physiotherapist-participants reflected on prehabilitation and suggested the information may be necessarily too vague because patients might experience such different symptoms (*Quote 44*). However, they did recognise that becoming accustomed to the exercises could be beneficial (*Quote 45*).

The use of video consultation was an unforeseen change to our GRRAND-F protocol as a result of the COVID-19 pandemic. Patient-participants and physiotherapist-participants reported both the benefits (accessibility) and drawbacks (lack of hands on assessment) of video consultations (*Quotes 46, 47, 48*).

The overall picture seems to be a desire for a hybrid delivery system, where patients can attend some face-to-face sessions, especially early in their rehabilitation, to achieve experiential learning combined with psychosocial reassurance. Then to have the opportunity to attend some sessions via video consultation, especially for those living further away. This could reduce the number of cancelled sessions and provide a more continuous support structure, which is patient-centred.

None of the patient-participants had sought any additional private medical care. They sought out information from friends and joined support groups for psychosocial support (*Quote 49*). This suggests their needs were met by the physiotherapy exercises and perhaps the delivery and support structure is an area which we need to target.

### Theme 5: Feasibility

We received universal agreement among our patient-participants and physiotherapist-participants that this research was needed because *'the unmet need in this population is so profound'*. Consequently, all were happy to be approached and offered the opportunity to participate (*Quotes 50 and 51*). The physiotherapist-participants shared a concern that they needed further administration support in order to scale up to an RCT (*Quote 52*).

#### Randomisation

Some patient-participants understood why the randomisation process allocated them to usual care or intervention. However, some members of the usual care group misinterpreted the reason for their allocation into usual care. One felt they did not 'qualify' (*Quote 53*) and another thought it was because he lived further away from the hospital (*Quote 54*).

Participants highlighted that randomisation occurs presurgery, which is a particularly stressful and emotional time and, therefore, it is very important that the research team offer time and compassion for study participants who are not allocated into the intervention arm (*Quote 54*).

#### Patient-reported outcome measures

The patient-reported outcome measures included shoulder pain and function indices and health-related quality of life measurements. Several patient-participants felt the measurements were lengthy and repetitive and others felt the mental health questions were *'a waste of time'* (*Quote 56*).

#### Recruitment

Patient-participants and physiotherapist-participants suggest that patients are likely to agree to participate in an RCT because there is very little support in usual care. Motivation to join research might be influenced by the patient's current dysfunction and physiotherapist-participants explained that this might not manifest until much later on in the rehabilitation journey (*Quote 57*).

## DISCUSSION
### Key findings

This study identified the overwhelming need for additional support, to augment the usual care available in the UK's National Health Service for patients with HNC undergoing ND surgery in order to improve their long-term outcomes. It also highlighted the importance of psychosocial support and integrating behaviour change techniques into complex interventions to support patients through their long-term rehabilitation journey.

Previous qualitative investigations reported the profound biopsychosocial impact of HNC and its management on patient's quality of life.[23 24] Our results supported earlier qualitative work with this population[18] and found that ND does not always, but can, have a large negative impact on the patient's physical and mental functioning which decreases a patient's quality of life.

The physiotherapy delivered in GRRAND encouraged tailored physical rehabilitation alongside psychosocial education and support, which is documented as an important element in caring for patients with cancer.[15] Some of this is imparted during inpatient care, however, the patients explained that when they are inpatients they could not take all the information in and they explained that because their symptoms evolved over time they might not realise the impact on their lives until much later after their surgery.

Evidence is growing to support the importance of patient-centred conversations in cancer rehabilitation, ensuring patients are equally empowered to ask questions and seek reassurances from their healthcare professionals.[16] Implementing a standardised methodology as part of the GRRAND intervention could assist in reducing social inequality in healthcare conversations. Treatment of patients with HNC varies due to the variety of clinical presentations. Prior evidence supports the integration of personalised strategies.[19] The evidence from this study suggests that alongside personalised strategies, the GRRAND intervention could try and encourage self-management and empowerment to explore and discuss more options with healthcare providers in order to support long-term, meaningful improvements.

Patients post-ND recognised long-term effects of their surgery on their mental and physical well-being. The GRRAND intervention was recognised as a useful tool which can generate long-term behavioural changes in patients to support their lifelong rehabilitation post-ND. However, recent evidence suggests[17 25] that in order to maximise the effectiveness of an intervention such as GRRAND, for HNC patient rehabilitation, it must implement behaviour change strategies identified from a behavioural diagnosis using the COM-B model of behaviour change.[22] Therefore, the findings from this study recommend the next iteration of the GRRAND intervention is guided by the COM-B model of behaviour change intervention development.

The behaviour change model could use findings which suggested it could be important to communicate the nature of the variation in symptomatology over time to increase patient motivation and engagement in a 'preventative' intervention framework.

The participants did not raise any strong problems regarding the feasibility of conducting an RCT to test the effectiveness of the GRRAND intervention according to the Consolidated Standards of Reporting Trials guidelines for pilot and feasibility testing.[26]

## Strengths and limitations

The key strength of this study is the novelty of these findings. An overarching message from the data was how overlooked this patient population has been and how complicated their physical rehabilitation journeys may be.

One researcher (BF) conducted all the interviews and conducted the framework analysis process; this led to the researcher having an immersive analytical experience. However, this does leave the analytical interpretation open to bias from a single perspective. We aimed to counterbalance this by performing iterative analysis presentation and discussion with the multidisciplinary trial management group (surgeons, physiotherapists, researcher analysts and patient perspective).

The patient-participants we interviewed had remained engaged with the GRRAND-F Study for over 6 months. These months were during the COVID-19 pandemic, which presented additional burden and strain. Therefore, this group might represent a subgroup of the patient population who are highly motivated by research.

The sample is small and while it meets theoretical requirements of including both men and women from both intervention and usual care experiences, its generalisability is limited. Due to COVID-19 interruption all the participants were from one geographical location rather than from the six locations we had anticipated. This sample is highly homogeneous (all white, all living in one area) and therefore limits the generalisability of these findings.

A problematic limitation was that four participants who withdrew from the study also declined to be interviewed. Consequently, these data and our interpretation did not include perspectives from those who did not engage with GRRAND. Therefore, our interpretation can only be applied to those who do engage with the GRRAND intervention and this may leave a group of our target population unrepresented.

However, by triangulating participant data with physiotherapist data we gain reflections from the physiotherapists who work across different geographical locations and with a broader more heterogenous patient population.

## CONCLUSIONS

The patient-participants and physiotherapist-participants perceived the discharge leaflet and physiotherapy intervention to be acceptable. This patient group experiences huge variation in their physical, mental and social dysfunction postsurgery. The dysfunction usually presents once the patients have been discharged from inpatient care and returned home to their daily lives. Sometimes the dysfunction can develop over months or years. The GRRAND intervention could be tested as a preventative intervention which aims to upskill patients with physical exercises and the physiotherapy sessions provide essential psychosocial support to encourage patients to engage in long-term proactive exercise behaviours.

**Contributors** AM managed the recruitment of participants. BF interviewed participants, analysed the data and wrote the manuscript and is responsible for overall content as the guarantor. TS, SL, AM and SW were all involved with iterative secondary analysis and reviewed the manuscript.

**Funding** This work was supported by the National Institute of Health Research grant number PB-PG-1217-20031.

**Competing interests** None declared.

**Patient and public involvement** Patients and/or the public were involved in the design, or conduct, or reporting, or dissemination plans of this research. Refer to the Methods section for further details.

**Patient consent for publication** Consent obtained directly from patient(s)

**Ethics approval** This study involves human participants and was approved by the Research Ethics Committee South Central (Oxford B) 29 October 2019 (National Research Ethics Committee Number: 19/SC/0457). Participants gave informed consent to participate in the study before taking part.

**Provenance and peer review** Not commissioned; externally peer reviewed.

**Data availability statement** All data relevant to the study are included in the article or uploaded as supplementary information.

**ORCID iD**
Beth Fordham http://orcid.org/0000-0001-5996-3563

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
