## [Reviewer comments · BMJ Open]

ARTICLE DETAILS

TITLE (PROVISIONAL)	Patient and physiotherapist perceptions of the Getting Recovery Right After Neck Dissection (GRRAND) rehabilitation intervention: A qualitative interview study embedded within a feasibility trial.
AUTHORS	Fordham, Beth; Smith, Toby; Lamb, Sarah; Morris, Alana; Winter, Stuart

VERSION 1 – REVIEW

REVIEWER	wu, Peixia Eye & ENT Hospital of Fudan University
REVIEW RETURNED	09-Jun-2022

GENERAL COMMENTS	Thank you for letting me review the manuscript entitled Patient and physiotherapist perceptions of the Getting Recovery Right After Neck Dissection (GRRAND) rehabilitation intervention: A qualitative interview study embedded within a feasibility trial. This paper reported Patient and physiotherapist perceptions of the GRRAND, and explored the possibility of conducting an RCT. The finding is helpful for clinicians to make support decisions. The biggest concern of this manuscript is that 4 people who withdrew from the study refused the interview, and the data may be insufficient in terms of acceptability, adherence, feasibility, and outcomes reported in the results. This point should be addressed in the limitation part. For the discussion section, the author described the overwhelming need for additional support to be provided to HNC patients, but they did not discuss what did this mean to GRRAN and what should we do to modify or redesign the rehab program. Some minor concerns Could you pls tell the readers what has already been known based on previous studies and the gap between research? P10L3 typos " capable of doing them do it"; P15L6 "they symptoms evolve"?, pls check.
---

REVIEWER	Gane, Elise University of Queensland, School of Health and Rehabilitation Sciences
REVIEW RETURNED	15-Jun-2022

GENERAL COMMENTS	Thank you for the opportunity to review this qualitative study embedded within the GRRAND pilot trial. Conducting such a study within a trial, with patients and therapists who delivered the intervention, is good practice and I commend the researchers for undertaking this step in their program of research.
---

	The COVID-19 pandemic has influenced the recruitment into this study, however the use of 8 patients and 5 therapists is appropriate for a pilot study. Four comments are offered for the authors to consider:  1. Methods section, when discussing the interviewer (health psychology researcher BF): could you please outline the relationship between BF and the research team. Was BF a member of the research team from study conception? The manuscript states “BP explained that she was acting independently of the physiotherapists and other trial members”. In what way did BF act independently? 2. There are several examples of incorrect grammar through the manuscript e.g. Results, theme 2, “they believed they were capable of doing them do it”. The authors should read the manuscript carefully for grammar and spelling please. 3. Discussion: “GRRAND encouraged correct physical rehabilitation” – ‘correct’ feels like an odd word choice here. Please reconsider. 4. The Discussion in general is very short, focussed on key findings, strengths and limitations, and conclusions. This article has been submitted as original research, not a short report, therefore the authors are encouraged to 'flesh out' the discussion and make reference to previous publications that have reported qual findings from pilot RCTs, perhaps of exercise interventions in patients with cancer in general. Place these results in the context of previous research.
--	---

VERSION 1 – AUTHOR RESPONSE

Reviewer: 1

Dr. Peixia wu, Eye & ENT Hospital of Fudan University

Comments to the Author:

Thank you for letting me review the manuscript entitled Patient and physiotherapist perceptions of the Getting Recovery Right After Neck Dissection (GRRAND) rehabilitation intervention: A qualitative interview study embedded within a feasibility trial. This paper reported Patient and physiotherapist perceptions of the GRRAND, and explored the possibility of conducting an RCT. The finding is helpful for clinicians to make support decisions.

- Thank you for your time and care in reviewing this manuscript. We really hope it is a useful step for supporting clinicians in making decisions.

- The biggest concern of this manuscript is that 4 people who withdrew from the study refused the interview, and the data may be insufficient in terms of acceptability, adherence, feasibility, and outcomes reported in the results. This point should be addressed in the limitation part.

- Thank you, we fully agree with this important point and I have edited the discussion to make this point fully transparent as a limitation to this study.

For the discussion section, the author described the overwhelming need for additional support to be provided to HNC patients, but they did not discuss what did this mean to GRRAN and what should we do to modify or redesign the rehab program.

- I absolutely agree we needed to make the next step of applying these findings to how we will improve and redesign the intervention/design for the next phase. I have added an explanation of how

we intend to perform behavioural diagnosis and identify tailored behaviour change technique to integrate into the GRRAND intervention in order to increase adherence to GRRAND physiotherapy exercises. We also intend to target self-management and empowerment for the patients to ask for more help.

Some minor concerns

Could you pls tell the readers what has already been known based on previous studies and the gap between research?

- I completely agree this is an omission. I have added many more references in order to contextualise the current study data.

P10L3 typos "capable of doing them do it"; P15L6 "they symptoms evolve"?, pls check.

- As before, I really apologise for the lack of proof reading in this manuscript. I have done a full proof read and amended the specific typos identified.

Reviewer: 2

Dr. Elise Gane, University of Queensland

Comments to the Author:

Thank you for the opportunity to review this qualitative study embedded within the GRRAND pilot trial. Conducting such a study within a trial, with patients and therapists who delivered the intervention, is good practice and I commend the researchers for undertaking this step in their program of research.

The COVID-19 pandemic has influenced the recruitment into this study, however the use of 8 patients and 5 therapists is appropriate for a pilot study.

- Thank you for reviewing the manuscript for us. We were disappointed that COVID-19 caused our recruitment possibilities to shrink so drastically but we are heartened that you agree our sample is appropriate for a pilot study.

-

Four comments are offered for the authors to consider:

1. Methods section, when discussing the interviewer (health psychology researcher BF): could you please outline the relationship between BF and the research team. Was BF a member of the research team from study conception? The manuscript states "BP explained that she was acting independently of the physiotherapists and other trial members". In what way did BF act independently?

- I agree this is confusing. I have tried to aid transparency but explaining that BF was acting as an independent interviewer, she was not one of the physiotherapists, nor one of the trial designers, she was aiming to explain to the participants that she was purely interested in their responses and not trying to prove one hypothesis above another. I hope the edit has helped?

2. There are several examples of incorrect grammar through the manuscript e.g. Results, theme 2, "they believed they were capable of doing them do it". The authors should read the manuscript carefully for grammar and spelling please.

- As I mentioned above, I really apologise for the lack of proof reading in this manuscript, especially as you have taken the time to review it so carefully. I have read through and hope I have caught the typos you have identified and other outstanding ones.

3. Discussion: "GRRAND encouraged correct physical rehabilitation" – 'correct' feels like an odd word choice here. Please reconsider.

- Yes that does seem an odd word choice I have revised with "tailored" as that seemed less opinionated.

4. The Discussion in general is very short, focussed on key findings, strengths and limitations, and conclusions. This article has been submitted as original research, not a short report, therefore the authors are encouraged to 'flesh out' the discussion and make reference to previous publications that have reported qual findings from pilot RCTs, perhaps of exercise interventions in patients with cancer in general. Place these results in the context of previous research.

- I completely agree and have added references to contextualise the research in the current evidence base of this field.

Best wishes

VERSION 2 – REVIEW

REVIEWER	wu, Peixia Eye & ENT Hospital of Fudan University
REVIEW RETURNED	02-Aug-2022
GENERAL COMMENTS	Thanks for the opportunity to review the resubmission. I think the author has addressed my concerns. The resubmitted version is well-written and keeps to the point. I think it is ready to be accepted.